# Arterial Stiffness Assessment by Pulse Wave Velocity in Patients with Metabolic Syndrome and Its Components: Is It a Useful Tool in Clinical Practice?

**DOI:** 10.3390/ijerph191610368

**Published:** 2022-08-19

**Authors:** Monika Starzak, Agata Stanek, Grzegorz K. Jakubiak, Armand Cholewka, Grzegorz Cieślar

**Affiliations:** 1Department and Clinic of Internal Medicine, Angiology, and Physical Medicine, Specialistic Hospital No. 2 in Bytom, Batorego 15 St., 41-902 Bytom, Poland; 2Department and Clinic of Internal Medicine, Angiology, and Physical Medicine, Faculty of Medical Sciences in Zabrze, Medical University of Silesia, Batorego 15 St., 41-902 Bytom, Poland; 3Faculty of Science and Technology, University of Silesia, Bankowa 12 St., 40-007 Katowice, Poland

**Keywords:** metabolic syndrome, arterial stiffness, pulse wave velocity, insulin resistance, type 2 diabetes mellitus, dyslipidemia

## Abstract

Metabolic syndrome (MS) is not a single disease but a cluster of metabolic disorders associated with increased risk for development of diabetes mellitus and its complications. Currently, the definition of MS published in 2009 is widely used, but there are more versions of the diagnostic criteria, making it difficult to conduct scientific discourse in this area. Increased arterial stiffness (AS) can predict the development of cardiovascular disease both in the general population and in patients with MS. Pulse wave velocity (PWV), as a standard method to assess AS, may point out subclinical organ damage in patients with hypertension. The decrease in PWV level during antihypertensive therapy can identify a group of patients with better outcomes independently of their reduction in blood pressure. The adverse effect of metabolic disturbances on arterial function can be offset by an adequate program of exercises, which includes mainly aerobic physical training. Non-insulin-based insulin resistance index can predict AS due to a strong positive correlation with PWV. The purpose of this paper is to present the results of the review of the literature concerning the relationship between MS and its components, and AS assessed by PWV, including clinical usefulness of PWV measurement in patients with MS and its components.

## 1. Introduction

### 1.1. Metabolic Syndrome

Metabolic syndrome (MS) is not a single disease but a cluster of metabolic disorders such as abdominal obesity, atherogenic dyslipidemia, elevated blood pressure (BP), insulin resistance (IR), and higher glucose levels [1]. For decades, there has been a debate on the proper definition and diagnostic criteria of MS. There are different versions of the criteria for diagnosing MS, often taking into account different parameters, which may hinder the comparability of the research results obtained by different authors and make it difficult to conduct scientific discourse [2,3,4,5,6,7,8,9,10]. The International Diabetes Federation (IDF) issued a consensus in accordance with the American Heart Association (AHA) and the National Heart, Lung and Blood Institute (NHLBI) in 2009. According to the mentioned consensus, MS can be diagnosed if at least three of the five conditions in Table 1 are met [11].

MS is linked to an increased risk of developing type 2 diabetes mellitus (T2DM) and its complications. Cardiovascular diseases (CVDs) are one of the most important causes of morbidity and mortality in the population of patients living with T2DM [12,13,14]. Moreover, T2DM increases the risk of restenosis, which diminishes the efficacy of endovascular treatment of atherosclerotic CVD, worsens prognosis and may lead to the necessity of reintervention [15].

MS was shown to be associated with increased oxidative stress. Obesity and IR are the MS component that contributes the most to this relationship [16]. On the other hand, the role of oxidative stress in the pathogenesis of CVD is well established [17]. Although achieving a target low-density lipoprotein cholesterol (LDL-C) concentration adequate to the patient’s cardiovascular risk is one of the most important therapeutic goals in patients with atherosclerotic CVD [18], it should be noted that lipoproteins may be modified, especially by pro-oxidative factors that make them dysfunctional and more atherogenic, which means that the total concentration of individual plasma lipid fractions does not provide full knowledge about the state of the patient’s lipid metabolism [19].

Therefore, it is important to identify subclinical vascular damage in patients with MS and its components who should be taken into careful consideration for appropriate treatment. Increased arterial stiffness (AS) can be considered to be a good predictor of the development of subclinical cardiovascular dysfunction, so the assessment of AS can be useful in the identification of patients with increased cardiovascular risk [20,21].

### 1.2. Assessment of Arterial Stiffness

According to the American Heart Association as well as European experts’ consensus, the measurement of pulse wave velocity (PWV) remains a standard method to assess AS [22,23]. There are different devices applicable to the measurement of PWV, based on such methods as tonometry, oscillometry, ultrasonography, and magnetic resonance imaging [24].

However, there are some limitations to the measurement of PWV. Specialized equipment and experienced personnel are required to perform the measurement. In addition, care should be taken to ensure that the measurement is carried out in appropriate conditions, such as, among others: a quiet, dry room, ensuring thermal comfort; moreover, the measurement should be carried out after a ten-minute rest in a lying position. The BP value and heart rate (HR) should also be taken into account during the interpretation of the PWV value, as these parameters can influence the PWV measurement result [22].

### 1.3. The Purpose of This Paper

The purpose of this paper is to present the results of the review of the literature concerning the relationship between MS, its components, and AS, with the greatest emphasis on AS assessed by PWV, including the clinical usefulness of PWV measurement in patients with MS and its components.

## 2. Impact of Metabolic Syndrome Components on Arterial Stiffness

### 2.1. Elevated Blood Pressure

Increased systolic blood pressure (SBP) or diastolic blood pressure (DBP) are both MS components. In patients with high cardiovascular risk such as patients with MS, it is crucial to assess subclinical dysfunction of the cardiovascular system in order to initiate preventive measures, adequate life changes, and treatment early [25,26]. There are studies, both in humans and animals, that suggest that AS can precede high BP [27,28,29,30].

Increasing AS defines the loss of the arterial wall’s resistance to expansion by an increment in volume; as a result, it is linked to increased BP [31,32]. AS is closely related to aging, as the aorta and major arteries lose elasticity. As compensation, they undergo dilatation, which results in widened pulse pressure [33,34]. The effectiveness of this compensative mechanism is, however, limited by the greater arterial stiffening during systole in the large stiffer artery. The gold standard for measuring AS remains the velocity of arterial pressure waves, despite their limitation and dependence on age, BP, and HR [35,36,37]. Moreover, elevated serum leptin levels may influence the potential mechanism leading to sympathetic activation and therefore increasing PWV and elevated BP [38].

In a cohort study in which 54,849 middle-aged patients participated, it has been shown that both SBP and DBP depend on AS. Compared to patients with normal PWV values, moderate, moderate–high, and high PWV values were responsible for a 3.03-, 5.44-, and 7.87-times higher incidence of hypertension, respectively [39]. Hypertension was associated with a higher carotid–femoral PWV, regardless of sex, age, ankle–brachial index (ABI), body mass index (BMI), walking capacity, and HR (ß = 2.59 ± 0.76 m/s, b = 0.318, *p* = 0.003) [40].

However, in a four-week observational study performed by Kubalski et al., it was found that the PWV value changes with BP and HR. PWV was related to BP (*p* < 0.001) and HR (*p* = 0.07) levels between visits during the period [41].

In the Systolic Blood Pressure Intervention Trial (SPRINT), an analysis of 8450 patients demonstrated a 42% lower risk of death associated with a decrease in the PWV value observed after one year of antihypertensive therapy compared to patients in whom the PWV value remained not reduced. Furthermore, the results of the SPRINT study were independent of BP reduction and Framingham risk score [42].

In the management of arterial hypertension, it is crucial to pay attention to a low sodium diet, as well as exercise training. High sodium intake is related to water retention, an increase in systemic peripheral resistance, and changes in the structure and function of large elastic arteries [43]. The important role of a low sodium diet as a part of hypertension treatment has been proved [44,45]. Moreover, a meta-analysis published by Lopes et al. supports that exercise intervention based on aerobic, combined, or isometric exercise is suitable to improve PWV in adults with arterial hypertension [46]. According to the results published by Vamvakis et al., lifestyle intervention consisting of exercise and diet is associated with improvement in AS in patients with essential hypertension [47].

### 2.2. Central Obesity

When considering abdominal obesity, it has been shown that in obese patients the PWV is significantly higher compared to normal healthy individuals, proving that abdominal obesity is a significant factor in the development of AS [48,49,50]. BMI, visceral fat thickness, and fat mass are the strongest body fat measures related to PWV [51,52,53]. However, few studies have analyzed the sequential time changes of AS in obese patients with MS but without diabetes mellitus, who have undergone weight reduction [54,55].

In a prospective study performed by Kae-Woei Liang, it has been proved that after a three-month weight reduction program, brachial–ankle PWV in patients with obesity and MS, compared to the healthy control group, decreased slightly. However, in those in whom weight regaining was observed in a sixty-month follow-up period, compared to the healthy control group that also experienced weight regaining, the brachial–ankle PWV increased by a significantly higher amount. Interestingly, brachial–ankle PWV at the 60th month after weight regaining in obese patients with MS was even worse than the baseline values, was associated only with SBP or DBP increments, and when compared to healthy individuals proves to be beyond the aging process. In addition, brachial–ankle PWV values after weight regaining were independent of body weight, high-sensitivity C-reactive protein (hs-CRP), or insulin resistance changes [56]. However, it is still unknown what impacts a decrease in PWV value after weight reduction the most; whether it depends on changes in adipokines and inflammatory markers or is due to weight reduction per se [57,58].

The weight loss program in obese patients is based on diet changes and regular exercise [59,60,61]. In a review article prepared by Saladini et al., it has been shown that in a large number of studies regular exercise training, in MS patients, is confirmed to provide benefits not only in reducing body weight, decreasing BP, and improving metabolic profile but also in reducing sympathetic activity which leads to better endothelial function and arterial elasticity. Aerobic exercise has been shown to decrease the PWV value in most published studies [62]. Combined resistance training and aerobic exercise were shown to improve AS, nitric oxide, and inflammatory and metabolic markers in obese adolescent girls [63].

### 2.3. Lipid Metabolism Disorders

Dyslipidemia is a metabolic disturbance associated with an impaired concentration of basic lipid parameters, such as blood level of total cholesterol (TC), low-density lipoprotein cholesterol (LDL-C), high-density lipoprotein cholesterol (HDL-C), triglycerides (TG), and/or the presence of dysfunctional lipoproteins in the blood [19].

The whole spectrum of possible underlying pathophysiological mechanisms of the influence of lipid profile on atherosclerosis has not yet been well established. Lipid disorder may lead to cardiovascular disease in different ways [64]. Elevated TG level has been shown to promote endothelial dysfunction by stimulating the expression of endothelial mediators, such as endothelin-1 [65,66]. An elevated triglyceride glucose index (TyG) was associated with a significantly increased risk of AS and nephric microvascular damage [67,68]. However, the relationship between elevated TG and AS is not completely clear [69].

In a prospective study involving 1934 patients, the correlation between elevated TG and AS was analyzed. The authors of this study define AS by cardio–ankle vascular index (CAVI) as independent of BP compared to brachial–ankle PWV. The elevated level of TG was shown to be associated with higher CAVI values (OR 1.607, 95% CI 1.063–2.429, *p* = 0.024). Moreover, the results of this study were independent of other MS components, age, gender, smoking status, LDL-C, and statin treatment [70].

In the analysis performed by Sang et al., TG values, as well as TG/HDL-C index, were documented to be associated with increased PWV and can indicate the high risk of progression of AS in healthy men [71].

In the retrospective study conducted by Alparslan Kilic, in which data from 1584 patients were analyzed, age (r = 0.931, *p* < 0.001), BMI (r = 0.166, *p* < 0.001), TC level (r = 0.139, *p* < 0.001), LDL-C level (r = 0.119, *p* < 0.001), TG level (r = 0.099, *p* < 0.001), non-HDL level (r = 0.140, *p* < 0.001), and TG/HDL-C ratio (r = 0.083, *p* = 0.001) was shown to have a significant impact on PWV values [72].

HDL is considered to be a lipoprotein with protective properties in the process of atherogenesis. It transports cholesterol back to the liver and has some antioxidant features [73,74]. Lower levels of HDL-C are considered to increase cardiovascular risk. However, studies focused on proving that increasing HDL-C levels decrease the risk of cardiovascular diseases are not corresponding. The results of these studies challenge the concept that raising HDL-C levels will uniformly translate into a reduction in cardiovascular risk [75].

In the study cited before isolated HDL-C plasma level was not correlated with PWV values. However, the TG/HDL-C index was shown to be positively correlated with PWV and could be considered an independent risk factor for AS development [74].

Plasma atherogenic index (AIP) is defined as a logarithm from the TG/HDL-C ratio [76]. In a retrospective study performed by Choudhary, data from 615 patients without antihypertensive and lipid-lowering treatment was analyzed. The analysis revealed that the highest tertile with a mean AIP of 0.15 had the highest PWV. Moreover, AIP was not associated with aortic or radial BP [77].

### 2.4. Impaired Carbohydrate Metabolism

AS increases in patients with MS and IR [78,79]. Prediabetes metabolic variables affect AS and accelerated arterial aging from an early age [80,81]. The clinical diagnosis of IR is useful for assessing the risk of T2DM [82]. When assessing the IR index there is a need for insulin level measurement, which is a limitation of such indexes due to their high cost and variability depending on the utilized technique [83]. Recently described non-insulin-based IR index, the metabolic score for insulin resistance (METS-IR), was developed with the aim to quantify peripheral insulin sensitivity. METS-IR is a function of such variables as glucose level, BMI as well as TG and HDL-C levels [84].

In a prospective study published in 2019, researchers proved the correlation between METS-IR and PWV, SBP, and DBP independently of age, sex, smoking status, and hypertensive therapy. METS-IR in 34% predicts PWV changes (β = 0.290, *p* < 0.001). In addition, they proved METS-IR to have the strongest correlation with AS over other non-insulin indexes, including TG/HDL-C ratio. When including METS-IR in the Framingham Hypertension Risk Prediction Model, a more precise hypertension incident predicament was observed [85].

## 3. Metabolic Syndrome and Pulse Wave Velocity

MS is a group of cardiometabolic disorders. AS may explain the increased risk of CVD in patients with MS [86]. In a study by Kangas et al., it has been documented that PWV was 16–17% higher in patients with MS compared to the healthy control group. Moreover, it has been noticed that the increase in cardiovascular risk in patients with MS is higher in women than in men [87]. The pathophysiology of this difference is not completely clear. It was suggested that differences in testosterone levels could be responsible for hemodynamic effects in men and women with MS [88].

In a large European multicentre study, a significant impact of MS on PWV was confirmed, although heterogeneous effects of MS components on AS parameters [89]. In a prospective study involving 3807 subjects, it was documented that patients with MS and AS assessed by the cardio–ankle vascular index (CAVI) have the worst prognosis, with the highest percentages of major adverse cardiovascular events such as myocardial infarction, stroke, or cardiovascular death compared to patients without MS or MS with normal range of CAVI index [90]. AS, defined by CAVI, has been reported to be a predictor of cardiovascular events, independent of conventional risk factors for atherosclerosis [91]. Badhwar has taken into account AS in central and peripheral arteries in terms of endothelial dysfunction in patients with MS. The central arteries were mainly influenced by endothelial dysfunction and therefore induced structural changes, although the peripheral arteries were shown to be affected by functional alterations [92].

Some efforts have been made to find the determinants of AS in patients with MS beyond PWV. A body shape index (ABSI) calculated from waist circumference, BMI, and body height has been confirmed to identify people with MS and increased AS [93]. Nedogoda et al. have documented carotid–femoral PWV to be a better parameter for early vascular aging estimation in patients with MS in comparison to clinical scales, Systematic COronary Risk Evaluation (SCORE) scale, QRESEARCH cardiovascular risk algorithm (QRISK-3) scale, and Framingham scale [94].

Patients diagnosed with arterial hypertension have a high prevalence of metabolic syndrome regardless of controlled or resistant hypertension [95]. Hypertensive patients with MS were shown to be older (57.9 ± 12.2 vs. 52.7 ± 14.1 years, *p* < 0.001) and to have higher PWV (9.0 ± 2.3 vs. 8.4 ± 2.1 m/s, *p* = 0.001) than those without MS [96]. Predicting the risk of cardiovascular events in MS patients is crucial. In a prospective study involving 2728 middle-aged patients with MS, the occurrence of at least one cardiovascular event was associated with higher mean BP, aortic PWV, aortic augmentation index, and carotid intima-media thickness (*p* < 0.05 for all variables) [97]. Chen et al. have analyzed a group of patients diagnosed with coronary artery disease (CAD) in terms of MS and AS as the risk factor for future cardiovascular events. In Kaplan–Meier analysis, MS and AS assessed by PWV in patients with CAD were shown to be risk factors for hospitalization (*p* = 0.005) or increased all-cause mortality (*p* = 0.002) [98].

The conclusions from selected studies on vascular stiffness in patients with MS are presented in Table 2.

## 4. Importance of Analyzed Research in Routine Clinical Practice

As shown in our analysis, there is no single pattern for elevated risk of CVD in patients with MS. MS, as a group of metabolic disorders, is strongly associated with subclinical vascular damage, marked by increased PWV value [99,100]. As proved in the cited literature as well as a study performed by Gagliardino, PWV is significantly impaired in patients with complete pictures of MS. Furthermore, the level of PWV is significantly increasing with the increasing number of MS components, supporting the need for careful research for the early diagnosis of CVD in patients with MS as a cost-effective preventive strategy [101,102]. Moreover, higher glucose levels in the prediabetic range and IR might lead to higher AS and concentric remodeling of the heart muscle [103].

Abdominal obesity is an important component of MS. The ambulatory AS index of the obese subjects is significantly higher than in the healthy controls [104,105,106]. Moreover, sensitivity analyses demonstrated that higher PWV is associated with waist circumference-to-BMI ratio and it remains significant after adjustment for HR, metabolic risk factors, and inflammatory markers [107,108]. It is crucial for obese patients who have lost weight to maintain their status, as weight regaining worsens AS with a significant increase in PWV, especially in patients with obesity and MS [56]. Moreover, it is important to pay particular attention to patients with MS and AS in whom the adverse effect of metabolic disturbances on arterial function can be offset by an adequate program of exercises, which includes mainly aerobic physical training [109,110].

It is important to measure PWV under stabilized BP and HR values, particularly in patients with newly diagnosed hypertension in whom detection of organ damage mediated by asymptomatic hypertension has an impact on risk stratification [111,112]. PWV is positively correlated with both SBP and DBP [113]. An increase in PWV is associated with a significantly higher risk of hypertension incidence [114]. However, treatment of MS components is guided by its component level and clinical vascular damage. There are no guidelines for the treatment of subclinical vascular damage based on and guided by PWV. Chronic renin-angiotensin blockade (olmesartan) has been documented to lower AS partly independently of the corresponding reduction in BP [115]. The strategy to prevent cardiovascular and renal events based on AS (SPARTE Study) has been shown to reduce office and ambulatory SBP and DBP, and prevent vascular aging, but not to reduce cardiovascular outcomes despite the higher intensity of treatment when there was PWV normalization-driven strategy treatment compared to BP-driven strategy [116]. However, as the study authors underline, it was the first study to focus on the impact of AS on cardiovascular outcomes. Furthermore, the main limitation of the study was the small number of patients and its limited capacity for clinical events.

The risk of increased AS depends on hyperlipidemia at young ages [117]. Dyslipidemia therapy guidelines are focused mainly on LDL-C lowering when considering cardiovascular outcomes. However, as shown in studies cited before, the TG/HDL-C index was shown to be positively correlated with PWV and could be considered an independent risk factor for the development of AS. Thus, even after reaching the recommended levels of LDL-C, there are still lipid disorders that may increase residual cardiovascular risk [118]. AIP demonstrates the relationship between atherogenic and preventive lipoproteins. The link between PWV and AIP supports the view that the calculation of AIP should be included in the everyday clinical evaluation of the risk of cardiovascular disease, especially due to the easy calculation of AIP from routine lipid profiles. It could be an effective predictive value to detect early vascular aging and subclinical atherosclerosis [119,120].

As shown in a study performed by Gong et al., AS became significant with increasing numbers of the metabolic components [101]. However, multiple regression analysis revealed that not only are MS components responsible for PWV level, but also age, sex, HR, serum uric acid level, and ferritin level were shown to significantly affect the relationship between inflammation and PWV [86,121,122,123]. Therefore, more studies are required to improve the interpretation of PWV in the treatment of MS patients. However, PVW provides additional useful information to guide the treatment of a patient with MS. It was shown that PWV in MS patients might indicate a group of patients that requires a detailed examination for CVD and an early life-changing strategy and treatment [124]. This proactive attitude would provide a cost-effective preventive strategy to avoid the negative impact of CVD on quality of life and healthcare systems due to their higher care costs. PWV has been shown to be a predictor of incident cardiovascular events and arterial calcification. However, PWV is not yet a routine evaluation in patients with a high risk of CVD.

Due to the strong correlation between PWV and METS-IR that extends beyond sex, age, smoking status, and hypertensive treatment, as shown in the analyzed studies, METS-IR could be treated as a subrogating of AS in routine primary care [125]. Furthermore, METS-IR has a significant value in the prediction of hypertension incidence [126].

## 5. Conclusions

MS and its components are a significant problem for healthcare systems worldwide. CVD is the leading cause of morbidity and mortality, and, on the other hand, MS predisposes the development of CVD. The identification of patients with features of subclinical dysfunction of the cardiovascular system in the population of patients with MS and its components is also very important in clinical practice, and PWV measurement is a valuable tool for this purpose.

AS defined by PWV is positively correlated with both SBP and DBP. It may indicate subclinical organ damage in patients with hypertension. In addition, the decrease in PWV level during hypertensive therapy can identify a group of patients with better outcomes independently from BP reduction.

Obese patients with MS who have lost and regained weight are predisposed to have a significant increase in PWV, above their baseline levels. The adverse effect of metabolic disturbances on arterial function can be offset by an adequate program of exercises, which includes mainly aerobic physical training.

The TG/HDL-C ratio demonstrates the ratio between the atherogenic and preventive lipoproteins. The increased TG/HDL-C levels ratio is positively correlated with increased PWV levels and is an independent risk factor for cardiometabolic disease, with a consecutive correlation that was not proved separately for TG and HDL-C levels.

IR affects AS and accelerates arterial aging. Non-insulin-based IR index, such as METS-IR, can predict arterial stiffens due to a strong positive correlation with PWV.

We have presented a heterogeneous group of studies focused on PWV in patients with MS and its components. It should be highlighted that it is important to take into account the level of cardiovascular risk in the population with MS. In addition, several modalities are available in the assessment of AS including recording the pulse waves by a tonometer transducer, standard BP cuff, doppler ultrasound, and magnetic resonance imaging (MRI). Most of the studies presented have used the tonometry transducer method. However, different points of measurement were taken into consideration, such as carotid, femoral, brachial, and ankle. In addition, the distance between the two surface sites and the time delay between the waveform is used to determine PWV, and it may be impacted by the obesity in measurements.

Although this subject is of great interest and the knowledge of the association between AS and MS and its components has increased significantly, further research is necessary, which perhaps will allow the use of the measurement of PWV rroutinely in the treatment of patients with MS. A novel approach and applications of existing drugs to specifically target pathways involved in modulating AS may provide further support to its broader assessment and treatment to improve cardiovascular outcomes.

## Figures and Tables

**Table 1 ijerph-19-10368-t001:** Components of metabolic syndrome (MS) and its diagnostic criteria. MS can be diagnosed when at least three conditions are met [11].

Central obesity	increased waist circumference (cut-off values for male and female gender differ between populations and countries)
Impaired carbohydrate metabolism	fasting venous blood glucose concentration ≥ 100 mg/dL or pharmacological treatment of diagnosed carbohydrate metabolism disorders
Impaired lipid metabolism	triglycerides blood level ≥ 150 mg/dL (1.7 mmol/L) or pharmacological treatment of this lipid disorder
	high-density lipoprotein cholesterol blood level < 40 mg/dL in men or < 50 mg/dL in women, or pharmacotherapy for this lipid disorder
Arterial hypertension	systolic blood pressure ≥ 130 mmHg or diastolic blood pressure ≥ 85 mmHg, or taking of antihypertensive drugs by a patient with diagnosed arterial hypertension

**Table 2 ijerph-19-10368-t002:** Characteristics of some selected included studies on the impact of metabolic syndrome components on arterial stiffness.

	Results
**Elevated blood pressure**	
Chen et al. (2021) [39]	Estimated pulse wave velocity (ePWV) was positively correlated with systolic blood pressure (SBP) and diastolic blood pressure (DBP). For every 1 cm/s increase in PWV, SBP, as well as DBP, increased by 5.60 and 2.12 mmHg, respectively.
**Abdominal obesity**	
Kim et al. (2021) [49]	PWV was shown to be significantly associated with a higher waist-to-hip ratio (WHR) [for >0.90 in men and >0.85 in women: odds ratio (OR) 1.23; 95% confidence interval (CI) 1.06–1.42; *p* = 0.005; for the highest tertile compared to the lowest tertile: OR 1.38; 95% CI 1.15–1.66; *p* < 0.001], and with higher visceral fat area (VFA) (for ≥100 cm^2^: OR 1.39; 95% CI 1.20–1.60; *p* < 0.001; for the highest tertile compared to the lowest tertile: OR 1.77; 95% CI 1.48–2.12; *p* < 0.001).
Stanek et al. (2021) [50]	Physical activity has a beneficial effect on perivascular adipose tissue (PVAT) function among obese patients by reducing oxidative stress and inflammatory state.
Vianna et al. (2019) [51]	In linear regression analysis, the highest regression coefficients with PWV were observed for body mass index (BMI) (r = 0.30; 95% CI 0.25–0.35), visceral fat thickness (r = 0.30; 95% CI 0.24–0.35), and fat mass (r = 0.30; 95% CI 0.24–0.35), even after controlling for potential confounders (sex, race, birth weight, family income, family education, and maternal smoking during pregnancy).
Liang et al. (2018) [56]	Brachial-ankle PWV (baPWV) decreased insignificantly after weight loss (*p* = 0.240), while weight regaining significantly increased baPWV [from 3rd month (1358 ± 168 cm/s) to the 60th month (1539 ± 264 cm/s), *p* < 0.001].
**Lipid disorders**	
Zhao et al. (2019) [67]	Increased triglyceride glucose index (TyG) was associated with a higher incidence of carotid-femoral PWV (cfPWV) > 10 m/s, baPWV > 1800 cm/s, ankle–brachial index (ABI) < 0.9, microalbuminuria, and chronic kidney disease.
Pavlovska et al. (2020) [70]	High trigliceryde (TG) levels were associated with a high cardio–ankle vascular index (CAVI), even after adjustment for other cardio-metabolic components, age, gender, smoking status, low-density lipoprotein cholesterol (LDL-C), and statin treatment (OR 1.607, 95% CI 1.063–2.429, *p* = 0.024).
Sang et al. (2021) [71]	The mean baPWV values increased from 1349 cm/s to 1410 cm/s and individuals increased/persisted with high baPWV (outcome 1). Among the subjects who had normal baseline baPWV, in 100 subjects elevated baPWV occurred after 4.1 years of follow-up (outcome 2). logTG (OR 1.64 [95% CI 1.14–2.37] for outcome 1; 1.89 [1.14–3.17] for outcome 2) and logTG/HDL-C (1.54 [1.15–2.10] for outcome 1; 1.60 [1.05–2.45] for outcome 2) were significantly associated with progression of AS after adjusting for confounders.
**Insulin-resistance state**	
Hill et al. (2021) [78]	Endothelial serum and glucocorticoid kinase 1 (SGK-1) may represent a point of convergence for insulin and aldosterone signaling in AS associated with obesity and metabolic syndrome.
Antonio-Villa et al. (2020) [81]	57% (Δ_E→MY_ 95% CI: 31.7–100.0) of the effect of insulin resistance (IR) on altered PWV analysis was mediated by visceral adipose tissue (VAT). Moreover, VAT acts as a mediator of the effect of IR on increased mean arterial pressure (Δ_E→MY_ 35.7%, 95% CI 23.8–59) and increased hypertension risk (Δ_E→MY_ 69.1%, 95% CI 46.1–78.8). The obtained results showed that visceral adiposity is a modifier of the effect of IR on altered vascular hemodynamics, increased blood pressure levels, and hypertension risk.
Adeva-Andany et al. (2019) [82]	Cross-sectional and prospective studies confirm that IR is associated with a subclinical vascular injury in patients with diabetes, independently of standard cardiovascular risk factors.

PWV—pulse wave velocity; SBP—systolic blood pressure; DBP—diastolic blood pressure; WHR—waist-to-hip ratio; OR—odds ratio; CI—confidence interval; VFA—visceral fat area; PVAT—perivascular adipose tissue; BMI—body mass index; baPWV—brachial–ankle pulse wave velocity; TyG—triglyceride-glucose index; cfPWV—carotid-femoral pulse wave velocity; ABI—ankle–brachial index; CAVI—cardio-ankle vascular index; TG—triglycerides; LDL-C—low-density lipoprotein cholesterol; HDL-C—high-density lipoproteins cholesterol; SGK-1—serum and glucocorticoid kinase 1; IR—insulin resistance; VAT—visceral adipose tissue; METS-IR—metabolic score for insulin resistance.

## Data Availability

We used PubMed and Web of Science to screen articles for this narrative review. We did not report any data.

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
