# Peer review of "Arterial Stiffness Assessment by Pulse Wave Velocity in Patients with Metabolic Syndrome and Its Components: Is It a Useful Tool in Clinical Practice?"

_ijerph, 2022, doi:10.3390/ijerph191610368_

Round 1
Reviewer 1 Report
Monika Starzak et al., in the manuscript entitled “Metabolic syndrome and arterial stiffness a way to assess subclinical vascular damage”, aimed to summarize available data on the metabolic syndrome (MetS) and arterial stiffness, including the underlying mechanisms and clinical implications. The main topic of the manuscript is very interesting and in focus of clinical research, however, the several topics are not described comprehensively or are not described at all (such as the underlying mechanisms). Therefore, the manuscript needs to be significantly improved and several issues need to be addressed more carefully. In the present form the manuscript cannot be accepted for the publication. I do hope that the authors will find helpful my comments below in order to improve their interesting manuscript.
Major comments:
1. The manuscript should be significantly improved and more focused (with more details discussed) of the specific aspects of the MetS. What would be a novelty of this review? Also, the authors should consider to change the title as they are discussing components of the MetS and not the MetS per se. Otherwise, they should add a brief discussion if there were subjects with diagnosed MetS. If it is true, why did the authors decide to address single components of the MetS?
2. Please state clearly what is known by today on the topic you are covering and give more details about the studies you are discussing (e.g. could the studies be compared between them, if they had the same design (such as inclusion/exclusion criteria, background, etc), was the same definition used to diagnose MetS, etc);
3. The authors stated the following: “Some molecular data on proposed mechanisms that could support protective role on the kidney…”, however, I do not see any discussion about proposed mechanisms. Also, in the introduction section, the authors stated that the underlying mechanisms will be discussed;
4. Similarly, the authors stated: “In patients managing high cardiovascular disease risk such as patients with MS, it is crucial to assess preclinical stages in order to…”, this is something confusing, please explain it better, which preclinical stages do you mean?
5. Please be careful when citing an article. For instance, after the following statement: “Moreover, Susana Lopes meta-analysis supports that exercise intervention based on aerobic….”, two articles are cited (by Susana Lopes and by Anastasios Vamvakis); in addition, I suggest to say “a meta-analysis by” or something similar instead of “Susana Lopes meta-analysis” (likewise, later it is stated: “The Francesca Saladini and Paola Palatini review has shown”);
6. Row 150, the authors stated: “In one ongoing,1934 patients prospective study…“, however, the study is completed, it is not ongoing;
7. Please check carefully English language, the are some grammar errors and some repetitions (e.g. rows 100 (in a 4-week observation) and 102 (visits during the 4 weeks period), and rows 278 (tonometer transducer) and 280 (the tonometry transducer method), respectively).
Reviewer 2 Report
The article under adjudication reviews the role of metabolic syndrome and arterial stiffness to assess cardiovascular damage. I must laud the bold efforts of the authors to pursue this vast evolving field of study and bring it under one cumulative review. But on the other hand, the article needs much improvements in terms of presentation, structure and information incorporated. I have the following fundamental comments that I would like to be clarified and the authors may consider them to improve their article.
1) The title of the review is misleading. Although it addresses a broad topic, however, the review narrows down to only 4 topics (per Fig 2 selection criteria). I recommend modifying the title so that it addresses clearly the content/outcome of the review article.
2) Kindly remove all references prior to 2012, unless absolutely necessary. The field of MS is changing very fast, and a review incorporating 10yr or older references is not solicited.
3) The review article seems to be packed with information in section 3. I am unable to comprehend any real mechanistic finding or a flow of information in the individual sections. Also, Age factor is missing which is stated in introduction Line 52. The overall recommendation is to follow a structure consistent with all sections and present the information better.
4) The necessity of Figure 1 is questionable at the end of the introduction. If the authors are keen to elaborate PWV (irrelevant to the review topic), kindly consider a separate section and a better figure. Also I believe an imp citation (PMID:34326391) is missing.
5) The most important Table 1 must be presented better. This review article tabulates information without any critical discussion at the end. I cannot find any mechanical insights and most of the information in table1 is discussed in section 3, making this table completely redundant.
6) The ending of a good review must include a critical discussion on the topic as a whole and the author's take on the topic. Currently section 6 highlights major literature findings only.
Overall, I urge the authors to completely restructure the article. It is a difficult endeavor to bring such a vast topic under one roof - which makes presentation of information in a structured fashion very important, rather than stacking literature findings. I do hope to see a better revised version on this very interesting topic.
Round 2
Reviewer 2 Report
I thank the authors for their kind consideration and thorough revision of the manuscript. The current version has definitely improved from the initial submission. I still have to recommend the following minor revision points that the authors can consider for further improvements of the manuscript.
1) Table 1 is a near exact reproduction of Table 1 from Ref 11. I request the authors to kindly modify in accordance to their clinical perspective and more importantly comment on the physiologic outcomes discussed there. Kindly add subheadings to the table.
2) On a similar note, Table 2 of current manuscript needs an elaborate clinical correlation/perspective interpretation of the data from the authors. This will definitely help understand the changes in the numbers as aptly reported from the pertinent literature.
3) In the conclusions section: lines 364-372, it is not quite clear how the current findings of the review are weighted upon. Overall, once again, I recommend a structured interpretation of the numbers/change in numbers from the authors perspective.
